# Changes in social environment due to the state of emergency and Go To campaign during the COVID-19 pandemic in Japan: An ecological study

Rie Kanamori[1], Yuta Kawakami[2], Shuko Nojiri[1,3]*, Satoshi Miyazawa[4], Manabu Kuroki[5], Yuji Nishizaki[1]

1 Clinical Translational Science, Juntendo University Graduate School of Medicine, Tokyo, Japan, 2 Department of Mathematics, Physics, Electrical Engineering and Computer Science, Graduate School of Engineering Science, Yokohama National University, Kanagawa, Japan, 3 Medical Technology Innovation Center, Juntendo University, Tokyo, Japan, 4 LocationMind Inc., Tokyo, Japan, 5 Faculty of Engineering, Yokohama National University, Kanagawa, Japan

* s-nojiri@juntendo.ac.jp

**Data Availability Statement:** All relevant data files are available from the Dryad repository (https://doi.org/10.5061/dryad.f1vhhmgzh).

## Abstract

### Background

During the coronavirus disease 2019 (COVID-19) pandemic in Japan, the state of emergency, as a public health measure to control the spread of COVID-19, and the Go To campaign, which included the Go To Travel and Go To Eat campaigns and was purposed to stimulate economic activities, were implemented. This study investigated the impact of these government policies on COVID-19 spread.

### Methods

This ecological study included all 47 prefectures in Japan as samples between February 3 and December 27, 2020. We used COVID-19 cases and mobility as variables. Additionally, places where social contacts could accrue, defined as restaurants, companies, transportation, and tourist spots; mean temperature and humidity; the number of inhabitants in their twenties to fifties; and the number of COVID-19 cases in the previous period, which were factors or covariates in the graphical modeling analysis, were divided into five periods according to the timing of the implementation of the state of emergency and Go To campaign.

### Results

Graphical changes occurred throughout all five periods of COVID-19. During the state of emergency (period 2), a correlation between COVID-19 cases and those before the state of emergency (period 1) was observed, although this correlation was not significant in the period after the state of emergency was lifted (period 3). During the implementation of Go To Travel and the Go To Eat campaigns (period 5), the number of places where social

**Funding:** The authors received no specific funding for this work.

**Competing interests:** Kanamori is an employee of Sanofi, K.K., Japan. Miyazawa is an employee and shareholder of LocationMind Inc., Tokyo, Japan.

contacts could accrue was correlated with COVID-19 cases, with complex associations and mobility.

## Conclusions

This study confirms that the state of emergency affected the control of COVID-19 spread and that the Go To campaign led to increased COVID-19 cases due to increased mobility by changing behavior in the social environment where social contacts potentially accrue.

## Introduction

Coronavirus disease 2019 (COVID-19) originated in Wuhan, China [1]. From there, it spread rapidly worldwide and was declared a pandemic by the World Health Organization in March 2020 [2]. At the beginning of February 2021, the number of infected people worldwide was >100 million [3]. In Japan, the first case was reported mid-January 2020, and the subsequent exponential increase in the number of cases led to 235,752 confirmed cases as of January 1, 2021 [4].

Because the main route of transmission of COVID-19 among humans is close contact and respiratory droplets [5,6] in not only symptomatic period but also asymptomatic and pre-symptomatic period [7], it is considered highly infectious. Thus, it is critical to reduce close contact. Many countries implemented public strategies, such as lockdowns and mobility restrictions, to stop the spread. After confirmation of the first wave of the COVID-19 outbreak, the Japanese government declared a national state of emergency, which was not a lockdown but rather a noncompulsory restriction (S1 Table), from April 6, 2020, for seven prefectures in metropolitan areas and from April 16, 2020, for the remaining prefectures across Japan [8]. They requested people to refrain from unnecessary outings, including going to restaurants, schools, and public facilities, and traveling to avoid the "Three Cs," namely, closed spaces, crowded spaces, and close-contact settings. After the restrictions were lifted on May 25, 2020, the number of infected cases decreased temporarily [9]. Controversially, from the end of July 2020, the Japanese government conducted the Go To campaign as an economic measure to stimulate economic activities, which contributed to increasing social contacts and human mobility in Japan. The campaign included Go To Travel, from August 2020, which encouraged all residents to travel around Japan by providing discounts for travel expenses and coupons, and Go To Eat, from October 2020, which encouraged people to eat out at restaurants and bars by providing points for discounts [10]. When the Go To campaign had just started, the second COVID-19 wave occurred. Because the third big wave occurred during the New Year holidays, the campaign was suspended at the end of 2020, and the state of emergency was declared again at the beginning of January 2021.

Social distancing measure, stay-at-home orders and lockdowns implemented as non-pharmaceutical interventions in some countries contributed to the reduction of mobility [11–13], including domestic travel and outings to neighboring places (such as workplaces, restaurants, bars, and schools), which was associated with the mitigation of the short-term spread of the disease in previous studies [14–19]. However, some studies reported that mobility and social contacts increased substantially after lifting mobility restrictions and reopening economic activities [20,21]. Regarding environmental factors related to increased social contacts and human mobility in daily life, facilities such as restaurants, company premises, public transportation, hotels, and leisure spots are closed places where the likelihood of transmission is

considered potentially high [22–24]. Thus, after lifting restrictions and reopening economic activities, mobility and social contacts would increase at these specific facilities and lead to rebound of transmission depending on the level of mitigation [25]. Weather conditions are also potential factors that influence human activity [26,27].

The impact of the state of emergency in Japan on controlling the spread of COVID-19 through mobility restrictions, various environmental factors that could have affected the spread of COVID-19 throughout 2020, and the implementation of the Go To campaign in the middle of the pandemic, which might have made the context more complicated, have not been investigated. Thus, this study aimed to clarify how the execution of the state of emergency and the Go To campaign impacted the spread of COVID-19 in Japan. We used graphical modeling, taking into account mobility, which is related to the number of social contacts in daily life, and climate based on data for five time periods related to changes in government policies.

## Materials and methods

### Study design

This ecological study was approved by the ethics committee of Juntendo University and was conducted using publicly available data. The study period was from the early stage of the outbreak to the end of 2020, when the Go To campaign was suspended. To verify the impact of the state of emergency and Go To campaign, we divided the study period into five stages, as follows: period 1, before the state of emergency was executed (February 3 to April 19, 2020); period 2, during the state of emergency (April 20 to Jun 7, 2020); period 3, after the state of emergency was lifted (Jun 8 to August 9., 2020); period 4, after initiating the Go To Travel campaign (August 10 to October 18,2020); and period 5, after initiating the Go To Eat campaign (October 19 to December 27). Then, we compared the relationships among the variables in each period. We designated each period with adjustment for the International Standard for the representation of dates and times (ISO 8601) and approximately 14 days after the beginning and end of each measure (Fig 1). The incubation period for COVID-19 is estimated as on average 5–6 days up to 14 days [6,7,28,29]. We assumed reporting delay of polymerase chain reaction (PCR) test existed, which was reported around 3 days [30] last year because of an immature system of investigation, and people could take PCR tests after presenting symptoms in accordance with the guidance in Japan. We considered the sum of time including incubation period, testing and reporting, and we applied 14 days as lag between periods. To investigate the robustness of results of this study which set 14 days as lag time between periods, we conducted sensitivity analysis applying 0-day lag and 7-day lag. The units of this analysis are the prefectures of Japan ($N = 47$), and their demographics are shown in Table 1.

### Data source

We defined COVID-19 cases as cases of infection with severe acute respiratory syndrome coronavirus 2 that were newly confirmed as positive by PCR tests. The data of COVID-19 cases across the 47 prefectures from February 3 to December 27, 2020, were obtained from the Toyo Keizai Online "Coronavirus Disease (COVID-19) Situation Report in Japan" by Kazuki Ogiwara [31]. The passengers of the Diamond Princess cruise ship that docked in Kanagawa, travelers quarantined in airports, and returnees on government charter flights from Wuhan were excluded. We also obtained public data as follows: demographic data (age) from the Ministry of Internal Affairs and Communications (2020), meteorological data (temperature and humidity) from the Japan Meteorological Agency (2020), data on shops and facilities where social contacts could accrue from the TownPage database (2020) established by the Nippon Telegraph and Telephone Corporation (NTT, http://itp.ne.jp/), and human mobility data from

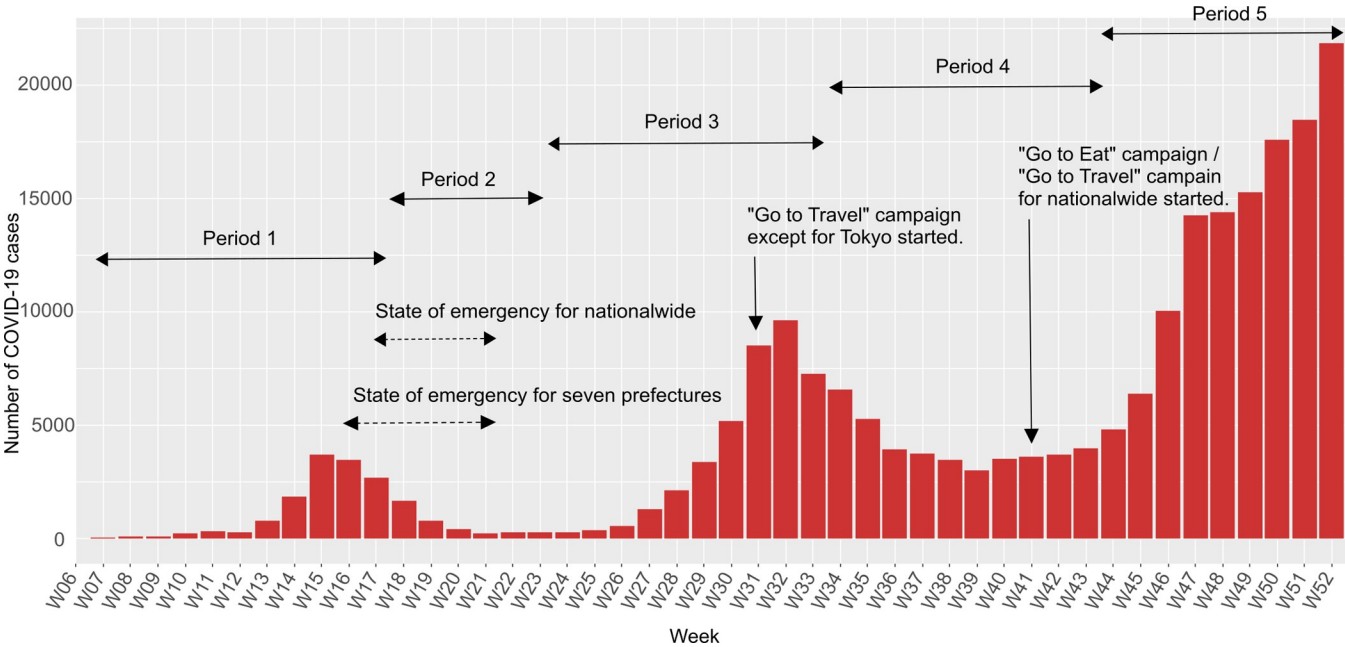

**Fig 1. Number of confirmed COVID-19 cases in Japan and government policies from February 3 to December 27, 2020.** The timing of the implementation of the state of emergency, Go To Travel campaign, Go To Eat campaign, and the five periods are shown.

LocationMind xPop. LocationMind xPop uses aggregated people flow data originally collected by NTT Docomo, Inc., through their application service "Docomo Map Navi" using only the cell phone's location data collected upon user consent to the service's auto GPS function, and then processed by NTT Docomo in entirety and statistically before being provided to LocationMind Inc. The original location data is GPS data (latitude, longitude) sent at a frequency of every 5 minutes at the shortest interval and does not include information that specifies individuals.

## Variables

We defined twelve variables (Table 2). Eight variables (COVID-19 cases, inhabitants in their twenties to fifties, mean temperature, mean humidity, mobility from urban areas, mobility from rural areas, mobility from prefectures with ordinance-designated cities, and mobility from Tokyo) were extracted from the data source for each period. The other four variables (number of restaurants, companies, transportation systems, and tourist spots) were commonly used in the analysis across all periods. These variables were obtained from the TownPage database (geographical maps of their distribution; S1 Fig) and were selected as potential environmental factors that were considered as hotspots of COVID-19 transmission by increased social contacts and mobility [32]. These variables were considered to reflect the sociographic characteristics of each prefecture in Japan. Therefore, we used meteorological variables, inhabitants in their twenties to fifties, and the variables obtained from the TownPage database as covariates. Because the spread of COVID-19 during the previous period was considered to affect the next period, we also considered this as a factor. We selected the population in their twenties to fifties as a demographic factor because these subjects were of working age and considered more active than other generations in Japan. In fact, the cumulative number of infected cases in this age group was larger than that in other age groups in Japan [33].

**Table 1. Baseline characteristics of the 47 prefectures.**

| Prefecture | Population | Population density (per km$^2$) | Total COVID-19 cases (per 100,000) | Inhabitants in their twenties to fifties (per 100,000) |
|---|---|---|---|---|
| Hokkaido | 5,267,762 | 66.5 | 245.7 | 468 |
| Aomori | 1,275,783 | 127.6 | 34.3 | 447 |
| Iwate | 1,235,517 | 79.4 | 30.7 | 443 |
| Miyagi | 2,292,385 | 314.8 | 91.2 | 490 |
| Akita | 985,416 | 81.8 | 12.7 | 418 |
| Yamagata | 1,082,296 | 114.2 | 34.1 | 436 |
| Fukushima | 1,881,981 | 132.8 | 47.6 | 456 |
| Ibaraki | 2,921,436 | 468.1 | 79.0 | 482 |
| Tochigi | 1,965,516 | 301.5 | 62.9 | 484 |
| Gunma | 1,969,439 | 302.8 | 111.1 | 477 |
| Saitama | 7,390,054 | 1,933.6 | 179.1 | 515 |
| Chiba | 6,319,772 | 1,217.9 | 165.9 | 509 |
| Tokyo | 13,834,925 | 6,367.8 | 408.8 | 569 |
| Kanagawa | 9,209,442 | 3,813.6 | 211.9 | 529 |
| Niigata | 2,236,042 | 174.8 | 22.6 | 453 |
| Toyama | 1,055,999 | 243.6 | 51.4 | 462 |
| Ishikawa | 1,139,612 | 270.0 | 90.7 | 474 |
| Fukui | 780,053 | 182.0 | 44.4 | 463 |
| Yamanashi | 826,579 | 180.6 | 62.9 | 468 |
| Nagano | 2,087,307 | 150.0 | 54.1 | 456 |
| Gifu | 2,032,490 | 185.9 | 101.5 | 470 |
| Shizuoka | 3,708,556 | 465.3 | 69.2 | 475 |
| Aichi | 7,575,530 | 1,457.8 | 206.9 | 518 |
| Mie | 1,813,859 | 306.1 | 67.5 | 477 |
| Shiga | 1,420,948 | 351.6 | 76.5 | 494 |
| Kyoto | 2,545,899 | 556.9 | 175.0 | 489 |
| Osaka | 8,849,635 | 4,627.8 | 326.9 | 512 |
| Hyogo | 5,549,568 | 647.4 | 168.0 | 487 |
| Nara | 1,353,837 | 358.4 | 134.4 | 464 |
| Wakayama | 954,258 | 193.5 | 62.5 | 451 |
| Tottori | 561,175 | 157.2 | 16.9 | 444 |
| Shimane | 679,324 | 99.4 | 30.2 | 426 |
| Okayama | 1,903,627 | 264.6 | 66.5 | 467 |
| Hiroshima | 2,826,858 | 329.6 | 108.2 | 478 |
| Yamaguchi | 1,369,882 | 219.5 | 39.1 | 434 |
| Tokushima | 742,505 | 173.9 | 26.3 | 446 |
| Kagawa | 981,280 | 505.6 | 29.3 | 460 |
| Ehime | 1,369,131 | 233.7 | 30.3 | 448 |
| Kochi | 709,230 | 97.1 | 89.4 | 431 |
| Fukuoka | 5,129,841 | 1,024.1 | 161.1 | 487 |
| Saga | 823,810 | 331.4 | 53.8 | 450 |
| Nagasaki | 1,350,769 | 317.3 | 43.4 | 435 |
| Kumamoto | 1,769,880 | 234.3 | 97.0 | 446 |
| Oita | 1,151,229 | 177.4 | 53.9 | 442 |
| Miyazaki | 1,095,903 | 137.5 | 65.2 | 433 |
| Kagoshima | 1,630,146 | 172.8 | 59.3 | 432 |

(*Continued*)

**Table 1.** (Continued)

| Prefecture | Population | Population density (per km²) | Total COVID-19 cases (per 100,000) | Inhabitants in their twenties to fifties (per 100,000) |
|---|---|---|---|---|
| **Okinawa** | 1,481,547 | 639.1 | 352.0 | 494 |

The data on the population, population density, and inhabitants in their twenties to fifties were obtained in 2020. The data of the total COVID-19 cases were accumulated over the study period, from February 3 to December 27, 2020.

We measured the volume of human mobility with a 7-day moving average and added inflow from other prefectures to the number of subjects in a prefecture(weekly changes shown in S3 Fig). Depending on the prefecture of origin, we categorized human mobility as either mobility from urban areas or and mobility from rural areas.

The population density of Tokyo was the highest, and the volume of human movement out of Tokyo was considered much higher than those from other prefectures (Table 1). Additionally, Tokyo had the highest number of infected cases among all the prefectures (Table 1). Therefore, we focused on human movement from Tokyo and conducted an influence analysis to clarify the impact on the spread of COVID-19 by excluding Tokyo from the analysis and adding two variables, namely, mobility from Tokyo to other prefectures and mobility from prefectures with ordinance-designated cities, instead of mobility from urban areas. We also calculated temperature and humidity using an averaged 7-day moving average for each day and averaged them for each period. We converted all variables, except mobility, into values per

**Table 2. Description of the variables.**

| Variable | Description |
|---|---|
| COVID-19 cases (per 100,000) | Newly confirmed SARS-CoV-2 cases by PCR tests |
| Inhabitants in their twenties to fifties (per 100,000) | Total number of inhabitants in their twenties to fifties |
| Restaurants (per 100,000) | Number of restaurants falling under the category of "Gourmet and Restaurants" in the NTT TownPage database, except for bento shops |
| Companies (per 100,000) | Number of companies falling under the category of "Business" in the NTT TownPage database |
| Transportations (per 100,000) | Number of railway stations, buses, ferries, and airports falling under the category of "Life" in the NTT TownPage database |
| Tourist spots (per 100,000) | Number of tourist information centers, rest stops, and hot springs falling under the category of "Travel and Accommodation" in the NTT TownPage database |
| Mean temperature (˚C) | Averaged temperature |
| Mean humidity (%) | Averaged relative humidity |
| Mobility from urban areas (people) | Total volume of human mobility from prefectures with ordinance-designated cities and from Tokyo to all prefectures except for the origin |
| Mobility from rural areas (people) | Total volume of human mobility from prefectures not categorized as urban areas to all prefectures except for the origin |
| Mobility from prefectures with ordinance-designated cities [a] (people) | Total volume of human mobility from prefectures with ordinance-designated cities to all prefectures except for the origin and Tokyo |
| Mobility from Tokyo [a] (people) | Total volume of human mobility from Tokyo to all prefectures except for Tokyo |

SARS-CoV-2, severe acute respiratory syndrome coronavirus 2; PCR, polymerase chain reaction, NTT, Nippon Telegraph and Telephone Corporation.

[a]Mobility from prefectures with ordinance-designated cities and mobility from Tokyo were used instead of mobility from urban areas in the influence analysis.

100,000 people in each prefecture and transformed all variables, except mean temperature, into natural logarithms to adjust for the normal distribution.

## Statistical analyses

We performed graphical modeling to analyze the association of COVID-19 spread with the variables in each period [34,35]. Graphical Modeling is one of powerful tools to visually analyze the conditional independence/dependence structure of the whole set of observed variables (twelve variables in the paper). The regression analysis is unsuitable to conduct such analysis because the conditional independence/dependence relationships between a response variable and covariates are a main interest of regression analysis, but those between covariates are not. In this analysis, considering what variables have edges with the infectiousness number indicates that the infectious number is the outcome variable, and the other variables, including mobility, are explanatory variables in the regression analysis. We categorized the variables into two groups, taking into account the time order: (1) the group of factors (number of restaurants, companies, transportation systems, and tourist spots; number of inhabitants in their twenties to fifties; mean temperature and humidity; and number of COVID-19 cases from the previous period applied for periods 2–5) and (2) the group of outcomes (number of COVID-19 cases and all variables of mobility). Then, we conducted recursive covariance selection for each group by means of backward elimination.

To verify the performance of the model in the selection, we applied the Goodness of Fit Index (GFI) and deviance. We determined the model when the model criteria GFI is small to some level. Detailed methodology is presented in the Supporting information (S1 Text).

The graphs were quantified by partial correlations. Statistical analyses were performed using JUSE-StatWorks/V5[36].

In this study, the graphs were interpreted in a joint distribution as (1) an undirected edge (-) between two variables, representing a correlation without order or (2) no undirected edge (-) between two variables, representing conditional independence given all other variables without directed edges. Then, we compared the structure of correlations among variables for each graph throughout the five periods, focusing on the following correlations: among the group of factors, between the group of factors and the group of outcomes, and among the group of outcomes. Finally, we verified their changes and assessed whether the government policies had impacted the spread of COVID-19.

## Results

### Main analysis

We confirmed 219,789 infectious cases of COVID-19 across Japan in our study cohort from February 3 to December 27, 2020. S2 Fig shows geographic maps with visual representations of changes in cumulative confirmed COVID-19 cases in each period. Fig 2 shows the estimated conditional independence structure in each period. The structure of the association between variables categorized in the group of factors (number of restaurants, companies, transportation systems, tourist spots, and inhabitants in their twenties to fifties) remained unchanged throughout the five periods. S2 Table shows the process of recursive covariance selection in graphical modeling of period 1 in main analysis. In period 1 (deviance = 16.79, degrees of freedom = 26, GFI = 0.96), inhabitants in their twenties to fifties was directly correlated with COVID-19, and strongly with tourist spots (a). Tourist spots were correlated with mobility from rural areas (b). It showed that COVID-19 was indirectly correlated with tourist spots and mobility from rural areas (Fig 2A). In period 2 (deviance = 24.47, degrees of freedom = 34, GFI = 0.93), the number of

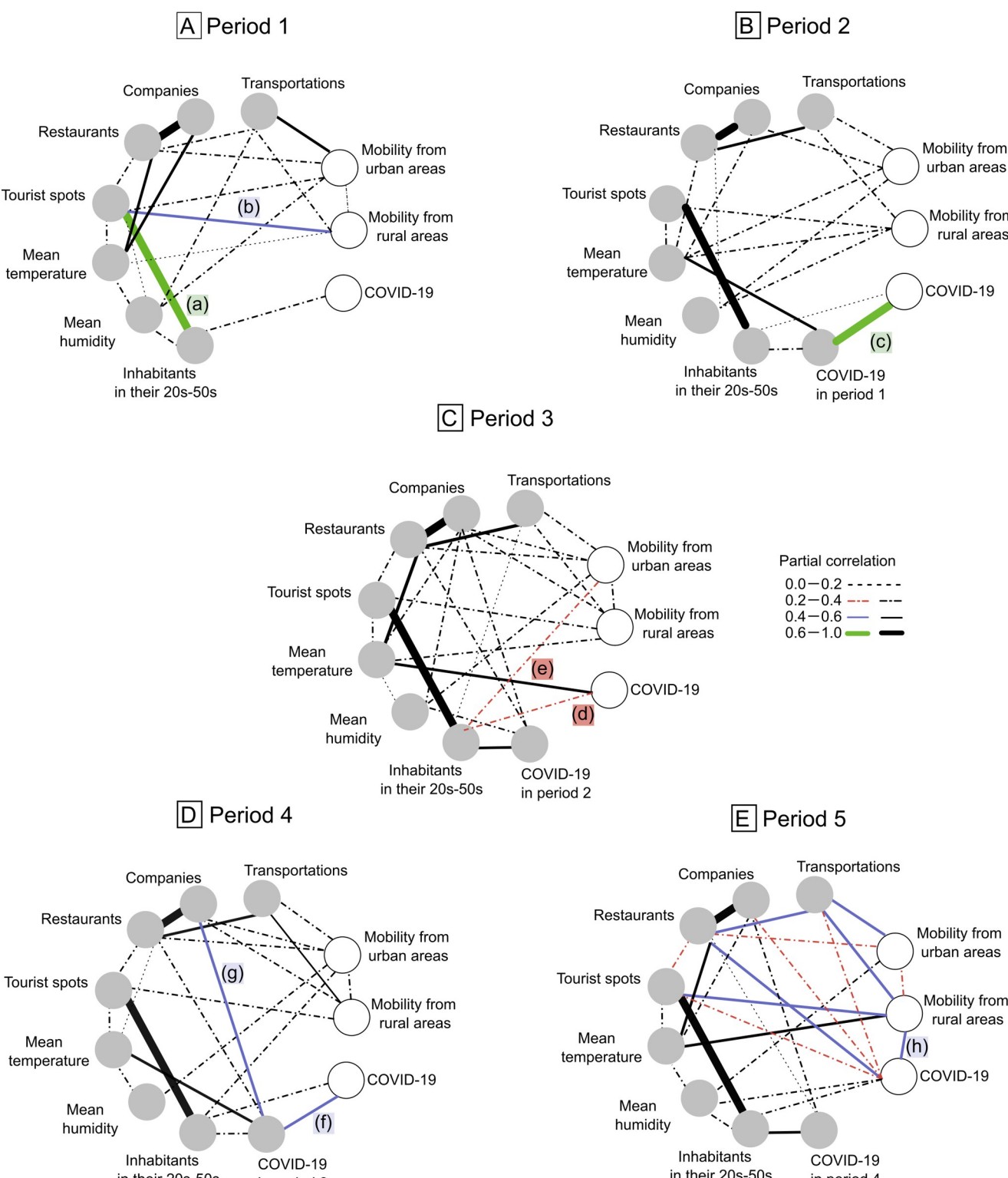

**Fig 2. Graph in each period of the main analysis using graphical modeling.** Fig 2A–2E show graphs of period 1, 2, 3, 4, and 5, respectively. The variables in the group of factors are shown as circles on a gray background, and variables in the group of outcomes are shown as circles on a white background. The directed edge (-) from the variables in the group of factors to the variables in the group of outcomes indicates the time order.

COVID-19 infections in period 1 was strongly correlated with COVID-19 in period 2 (c) (partial correlation = 0.65), (Fig 2B). In period 3 (deviance = 19.87, degrees of freedom = 29, GFI = 0.94), the direct correlation between COVID-19 in periods 2 and 3 was not significant. Inhabitants in their twenties to fifties were directly correlated with COVID-19 in period 3 (d) and with mobility from urban areas (e) (partial correlation = 0.39 and 0.23) (Fig 2C). It showed that COVID-19 was indirectly correlated with mobility from urban areas. In period 4 (deviance = 19.27, degrees of freedom = 33, GFI = 0.97), COVID-19 in period 4 was directly correlated with COVID-19 in period 3 (f) (partial correlation = 0.41). COVID-19 in period 3 was directly correlated with companies (g). It showed that companies were indirectly correlated with COVID-19 in period 3 (Fig 2D). In period 5 (deviance = 13.72, degrees of freedom = 26, GFI = 0.96), the direct correlation between COVID-19 in period 4 and COVID-19 in period 5 was not significant. COVID-19 in period 5 was strongly correlated with mobility from rural areas (h) (partial correlation = −0.43). The graph for period 5 was the most complicated. All variables related to increased numbers of social contacts (restaurants, transportations, companies, and tourist spots) were directly correlated with COVID-19 in period 5. The direct correlation of mobility from urban areas with transportations was stronger than previous period. Furthermore, the correlation of mobility from rural areas with transportations and with tourist spots were also stronger than the previous period (Fig 2E).

## Influence analysis

We aimed to clarify the impact of the mobility from Tokyo on the spread of COVID-19 by performing an influence analysis. The graphs were slightly complicated overall in comparison with the results of the main analysis (Fig 3 and S2 Text). Similar to the results of the main analysis, the correlation between COVID-19 in the present period and that in the previous period was only observed in periods 2 (a) and 4 (b), and COVID-19 in period 5 was strongly correlated with mobility from rural areas (c) (partial correlation = −0.41). Inhabitants in their twenties to fifties were correlated with mobility from Tokyo in all five periods (partial correlation = 0.45, 0.55, 0.48, 0.52, and 0.40, respectively) and with COVID-19 in the present period in periods 1,3–5 (partial correlation = 0.28, 0.23, 0.31, and 0.29, respectively) (Fig 3). Thus, mobility from Tokyo was not directly correlated with COVID-19 in the present period but indirectly correlated through inhabitants in their twenties to fifties. This relationship was not observed among other mobility variables, inhabitants in their twenties and fifties, and COVID-19 in the present period in period 2–5.

## Primary two changes in the graphs

Two primary changes were represented in the graphs throughout the five periods. First, in the period of the state of emergency (period 2; Fig 2B), a correlation of COVID-19 cases in the present period with that in the previous period was observed, although this correlation was not significant in the period after the state of emergency was lifted (period 3; Fig 2C). Second, in the period of the Go To Travel and Go To Eat campaigns (period 5; Fig 2E), this correlation was also not significant. The potential places related to increased social contact was correlated with COVID-19 in period 5, and mobility from rural areas was correlated with COVID-19 in period 5. Additionally, similar results were confirmed in the influence analysis (Fig 3). COVID-19 cases was indirectly correlated with mobility from urban areas in some periods of the main analysis and with mobility from Tokyo in all periods of the influence analysis through inhabitants in their twenties to fifties.

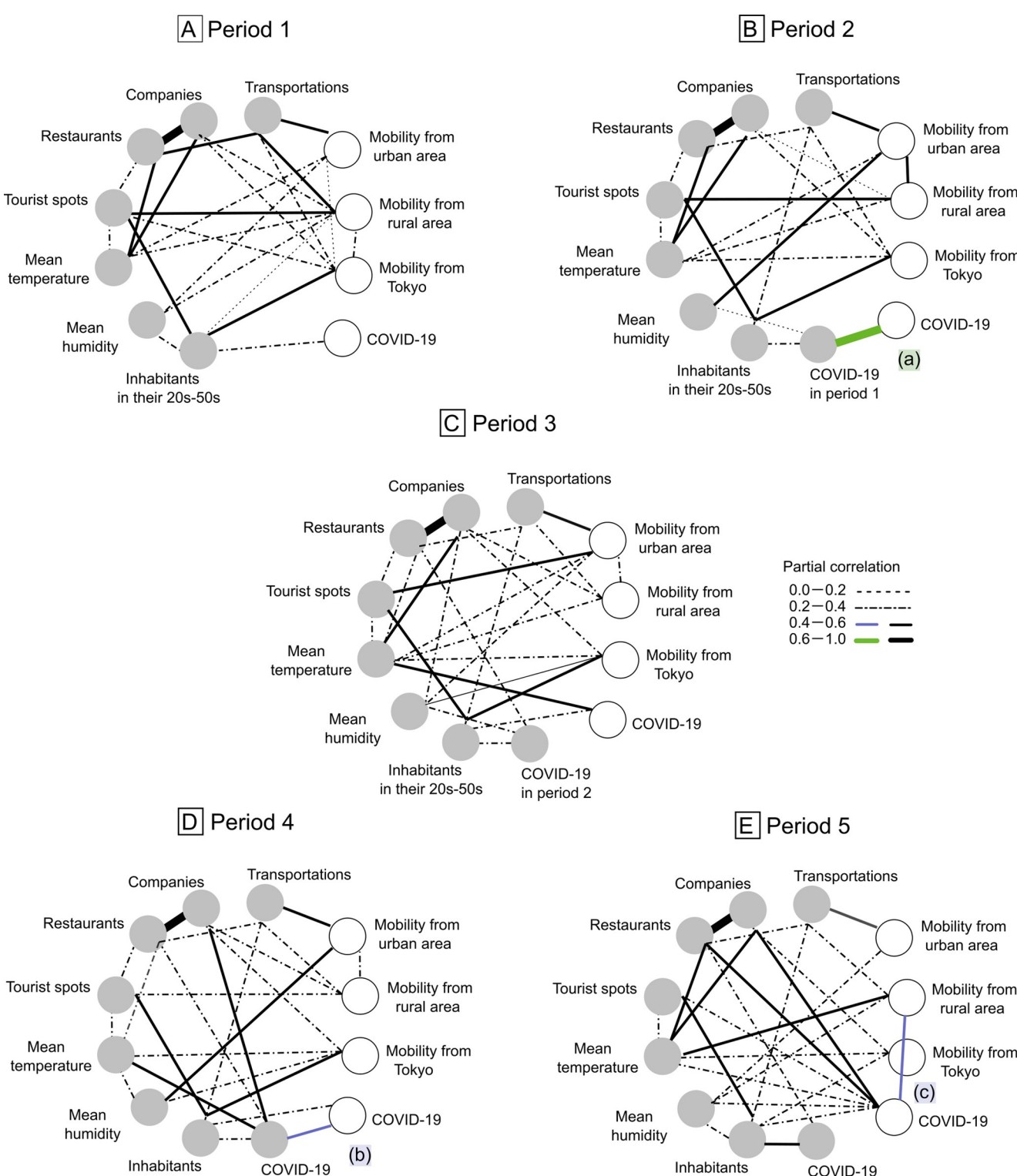

**Fig 3. Influence analysis graphs for each period using graphical modeling.** Fig 3A–3E show graphs of period 1, 2, 3, 4, and 5, respectively. The variables in the group of factors are shown as circles on a gray background, and variables in the group of outcomes are shown as circles on a white background. The undirected edge (-) from the variables in the group of factors to the variables in the group of outcomes indicates the time order.

### Sensitivity analysis

We performed sensitivity analysis with 0-day lag and 7-day lag (S2 Text, S3 Table, S4 and S5 Figs). The structure of each graph was not completely the same as that of main analysis through all periods, but the primary two changes in the graphs were consistent with those of main analysis.

## Discussion

Our study shows changes in the structure of association among the variables in the group of factors and those in the group of outcomes and changes in the entire structure of graphs throughout the five periods. The fixed structure of the association of variables (potential places related to the increase in the number of social contacts and inhabitants in their twenties to fifties) might show the structure where working generations congregate or interact in daily life. Based on this structure, the primary two graphical features and changes described in our results can be interpreted as follows.

First, the result that the strong association between COVID-19 cases in the present period and those in the previous period was not significant after the state of emergency was lifted (period 3) might be because people who contracted COVID-19 in the previous period did not affect the spread that was attributed to the effect of the state of emergency on controlling the spread of COVID-19, with a lag time. Second, the graph for period 5 was the most complicated because there were correlations among the variables related to increases in the number of social contacts, mobility, and COVID-19. Additionally, the strong association between mobility and COVID-19 in the present period was only observed in period 5. This suggests that the implementation of the Go To Travel and Go To Eat campaigns contributed to COVID-19 spread. Because the Go To campaign might have stimulated human mobility not only neighbor outings but also trips to other prefectures, the spread of infections might depend on increased mobility in the specific places such as transportations, tourist spots and restaurants. Accordingly, COVID-19 cases in the previous period (period 4) had little impact. Although the Go to Travel campaign started before period 5, campaign respondents increased considerably in October and November 2020 [37], which was within period 5.

A Japanese study also reported that travel-related increases of infectious COVID-19 cases were confirmed during the initial period of the Go To Travel campaign [38]. Although our study investigated the entire period of the execution of the campaign, the suggestion was consistent. Similarly, a previous study reported that the Eat Out to Help Out [39] scheme, which was implemented in the UK during the summer of 2020 as an economic measure to revitalize the food industry, might have caused an increase in new COVID-19 cases [40].

Furthermore, the result that mobility from Tokyo was indirectly correlated with COVID-19 through inhabitants in their twenties to fifties (Fig 3) imply that mobility from Tokyo to other prefectures especially in these generations could have had a greater indirect impact on COVID-19 cases than the mobility from other areas except for the period of the Go To Travel and Go To Eat campaigns (period 5), because this relationship was not observed in mobility from other areas.

Some studies that included Tokyo, Japan, have also suggested that behavioral changes related to eating out and traveling might have varied at the level of the individual [17] during the pandemic due to individual factors, such as social anxiety and dread, with self-restriction underlying an individual's level of risk perception [17,41–45]. In our study, variations among the models might also imply that changes in behavior were observed throughout the periods affected by the implementation of government policies.

## Findings

Changes in the conditional independence structure of variables in the five periods indicate that even though the state of emergency was a noncompulsory measure, it could contribute to the reduction of COVID-19 cases, and the Go To Travel and Go To Eat campaigns, which would increase mobility across Japan, might have led to increases in COVID-19 cases, thus influencing social environment and human behavior. Additionally, mobility from Tokyo could have had a greater impact on the spread of COVID-19 than mobility from other areas of Japan.

## Limitations

This study had some limitations. First, this was an ecological study, and ecological results shown at the aggregation level might differ from those obtained at the individual level, caused by ecological fallacy. Second, we did not take the intercorrelation among the samples (prefectures) into account when constructing the graphical model. Third, we did not calculate confirmed COVID-19 cases adjusted by the number of PCR tests per population of each prefecture and the difference in the implementation of PCR tests among regions. Fourth, the possible biases from the datasets exist in the following points; we applied TownPage datasets extracted in Aug 2020, however, the number of facilities might be slightly changed through the study period due to closing or new opening; and the mobility data is estimated volume calculated from GPS data. Fifth, regarding the assessment of the impact of the government policies on the control of the COVID-19 pandemic, we did not compare situations where the government policies were or were not implemented; therefore, we might have overestimated the impact. However, there is, in fact, no means to perform such an assessment.

## Conclusions

Our study suggests that the state of emergency reduced not only the influence of the number of COVID-19 cases in the prior period but also other variables, changing the social environment. Additionally, the implementation of economic measures that would increase social contacts and mobility during pandemics needs to be cautiously deliberated. As previous studies suggested, social distancing measures were considered still essential, especially in specific places right after lifting mobility restrictions and reopening economic activities during pandemic [25,46,47].

Although our study results may be as expected, this is the first study to demonstrate the effectiveness of considering various social and environmental factors to clarify whether public health and economic measures impacted the spread of COVID-19. Thus, our findings have identified important insights that should be deliberated to ensure optimal public health and economic measures in the future. Because the study period was limited to the first year of the COVID-19 pandemic, further research investigating the long-term data and mitigating the limitations of the present study are expected to contribute to a more efficient public health strategy.

## Supporting information

**S1 Fig. Geographical maps of total number of variables from The TownPage dataset: Restaurants, companies, transportations, and tourist spots in the 47 prefectures (as of Aug 2020).**
(TIF)

**S2 Fig. Geographical maps of the cumulative COVID-19 cases in the 47 prefectures in each period (February 3 to December 27, 2020).**
(TIF)

**S3 Fig. Weekly total volume of inter-prefecture mobility from February 3 to December 27, 2020.**
(TIF)

**S4 Fig. Graph in each period of the sensitivity analysis (0-day lag) using graphical modeling.**
(TIF)

**S5 Fig. Graph in each period of the sensitivity analysis (7-day lag) using graphical modeling.**
(TIF)

**S1 Table. Initial nationwide public health measures during the COVID-19 pandemic in Japan and other countries.**
(DOCX)

**S2 Table. Process of recursive covariance selection in graphical modeling of period 1 in main analysis.**
(PDF)

**S3 Table. The result of statistics in model selection in sensitivity analysis.**
(DOCX)

**S1 Text. Detailed methodology of recursive covariance selection in graphical modeling.**
(DOCX)

**S2 Text. Results of the influence analysis.**
(DOCX)

**S3 Text. The result of sensitivity analysis.**
(DOCX)

## Author Contributions

**Conceptualization:** Rie Kanamori, Yuta Kawakami, Shuko Nojiri, Manabu Kuroki, Yuji Nishizaki.

**Data curation:** Rie Kanamori, Yuta Kawakami.

**Formal analysis:** Rie Kanamori.

**Methodology:** Rie Kanamori, Yuta Kawakami, Shuko Nojiri, Manabu Kuroki.

**Resources:** Satoshi Miyazawa.

**Supervision:** Manabu Kuroki, Yuji Nishizaki.

**Visualization:** Rie Kanamori.

**Writing – original draft:** Rie Kanamori.

**Writing – review & editing:** Rie Kanamori, Yuta Kawakami, Shuko Nojiri, Satoshi Miyazawa, Manabu Kuroki, Yuji Nishizaki.

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
