## [Decision Letter · Decision Letter 0]

9 Nov 2021

PONE-D-21-31460Changes in social environment due to the state of emergency and Go To campaign during the COVID-19 pandemic in Japan: an ecological studyPLOS ONE

Dear Dr. Nojiri,

Thank you for submitting your manuscript to PLOS ONE. After careful consideration, we feel that it has merit but does not fully meet PLOS ONE’s publication criteria as it currently stands. Therefore, we invite you to submit a revised version of the manuscript that addresses the points raised during the review process.

Both referees identified significant shortcomings in the presentation of the methods, which need to be taken into account. Also, please consider the suggestion by Referee #2 about including an epidemiological modelling analysis to support the conclusions of the study.

We look forward to receiving your revised manuscript.

Kind regards,

Michele Tizzoni

Academic Editor

PLOS ONE

Journal Requirements:

3. PLOS requires an ORCID iD for the corresponding author in Editorial Manager on papers submitted after December 6th, 2016. Please ensure that you have an ORCID iD and that it is validated in Editorial Manager. To do this, go to ‘Update my Information’ (in the upper left-hand corner of the main menu), and click on the Fetch/Validate link next to the ORCID field. This will take you to the ORCID site and allow you to create a new iD or authenticate a pre-existing iD in Editorial Manager. Please see the following video for instructions on linking an ORCID iD to your Editorial Manager account: https://www.youtube.com/watch?v=_xcclfuvtxQ.

4. We note that Figure S1 in your submission contain [map/satellite] images which may be copyrighted. All PLOS content is published under the Creative Commons Attribution License (CC BY 4.0), which means that the manuscript, images, and Supporting Information files will be freely available online, and any third party is permitted to access, download, copy, distribute, and use these materials in any way, even commercially, with proper attribution. For these reasons, we cannot publish previously copyrighted maps or satellite images created using proprietary data, such as Google software (Google Maps, Street View, and Earth). For more information, see our copyright guidelines: http://journals.plos.org/plosone/s/licenses-and-copyright.

a) You may seek permission from the original copyright holder of Figure S1 to publish the content specifically under the CC BY 4.0 license.  

Reviewers' comments:

Reviewer's Responses to Questions

**Comments to the Author**

1. Is the manuscript technically sound, and do the data support the conclusions?

Reviewer #1: Partly

Reviewer #2: Partly

2. Has the statistical analysis been performed appropriately and rigorously? 

Reviewer #1: No

Reviewer #2: No

3. Have the authors made all data underlying the findings in their manuscript fully available?

Reviewer #1: No

Reviewer #2: No

4. Is the manuscript presented in an intelligible fashion and written in standard English?

Reviewer #1: Yes

Reviewer #2: Yes

5. Review Comments to the Author

Reviewer #1: This paper investigates the impact on COVID-19 spreading of the interventions implemented by the Japanese Government to stimulate the internal economy during 2020. Graphical modeling is used to understand the relationship between a set of variables (including mobility, socio-demographic factors, and weather information) and the number of cases reported in different prefectures of Japan. The methodology is applied to 5 different periods of 2020 to study the variation in the relationships between the features studied and the impact on COVID-19 spreading of the government interventions. The natural experiment provided by the implementation of the policies described is undoubtedly interesting and worth studying. Also the approach proposed by the authors is interesting and seems suitable to the purpose. Nonetheless, I think the paper needs substantial changes before being considered for publication. Below, I provide a list of the points that I think should be addressed by the authors. I will cite the line of the paper as L followed by the number of the line I’m referring to.

Major issues

1) The Introduction explains very well the interventions implemented by the Japanese government to stimulate the internal economy. However, I think a proper framing within existing literature is missing, apart from a couple of very general sentences. Previous influential studies dealt with similar problems (i.e., understand the role of certain points of interest such as restaurant etc in the spreading), such as:

https://www.nature.com/articles/s41586-020-2923-3

https://www.nature.com/articles/s41562-020-0931-9

On a more general basis, there is a vast literature on the impact of Non-pharmaceutical interventions (especially those impacting mobility) on COVID-19 spread, here a review that can help navigating it:

https://pubmed.ncbi.nlm.nih.gov/33612922/

More work have to be done to place the work in the existing literature

2) L103, authors state that the incubation period of COVID-19 is 14 days. This is wrong, also the references provided in the paper contradicts this point. Indeed, the average incubation period for COVID-19 (pre-Delta strains) is somewhere between 3 and 5 days. This value is used to define the 5 periods of the analysis, therefore this should be revised according to the right estimate.

Nonetheless, I am not sure if the incubation period is the right quantity to define the ‘lag’ between periods (by ‘lag’ I mean the number of days that are considered as a buffer between periods, and that is now set to 14 days). Indeed, there may also be a delay linked to the reporting of cases. For example, a person infected today will start to present symptoms after the incubation period but may be notified after because of delays associated with the surveillance system. Therefore, besides choosing an appropriate value I think it is essential that authors present as Supplementary Information some sensitivity analysis of the results for different lag values (what happens to the inferred graph if we introduce no lag? Are they stable to this change? What if the lag is equal to the incubation period? What if we also account for delays in reporting? Using relatively small lags should not affect the results much).

3) The definition “places where contacts could accrue” is vague even though it is a pretty important point for the paper. How are these places selected? I see that in Table 2 some categories of places are mentioned such as “Business” and “Life”. Why have some categories been chosen and not others? Within a given category, how do you choose some places and not others? The choice of places considered hotspots for COVID-19 transmission should be supported by evidence and references.

4) The paper lacks a proper materials and methods section. This undermines the overall understanding. First, the datasets used should be better explained:

- The TownPage dataset is used to define places where social contacts could accrue. Is it open access? Is it accurate? Is it complete? How is the dataset built? What are the categories listed and the type of places? Can you provide a plot showing the distribution of different points of interest in the prefectures? Please discuss

- The LocationMindxPro is used to measure mobility variation. Is it public? Where can the data be found? What is the size of the population over which mobility is computed? Can you show a plot with (even aggregate) mobility trends in Japan during the period of study? Do you work with raw data? If so, do you apply any preprocessing/inclusion criteria? Please discuss

- Also, what do you mean exactly by mobility? Do you assign each user a home location and see whether it is seen in other prefectures during the periods? Or do you define trips? If so, what is the duration of the trip? I think more detail on how you deal with mobility data is needed

Second, also the methods are not properly described:

- What is “recursive covariance selection for each group by means of backward elimination”? (L193) I understand this is the method used to infer the graphs, but how does it work? I think a few more details would help the reader to understand the methodology

- What are the Goodness of Fit Index (GFI) and deviance? (L194) How are they defined and what is their interpretation?

- L188, is the definition of the group of factors and of outcomes something that influences the graphical modeling or is it just a definition from the authors? Why is Mobility an outcome and not a factor influencing COVID-19 cases?

Minor issues

1) the very first lines (L45-50) of the paper needs some basic citations (the origin in Wuhan, the declaration of Pandemic by the WHO, the number of REPORTED cases by February 2021)

2) At L52 authors state that SARS-CoV-2 is highly infectious because of its route of transmission. This is not entirely true, it is highly infectious also because of other reasons, such as asymptomatic and pre-symptomatic transmissions

3) L142, how do you distinguish urban and rural areas?

4) L175, I do not understand the need to first compute a seven-days moving average for humidity and temperature and then an average over the all period. Why not directly average over the period?

5) L177, why is mobility not expressed per 100,000 as other variables?

6) L165, why add the number of inflow people to the people in the prefecture?

7) I find the results very difficult to read, now the section is just a long list of sentences like this one “inhabitants in their twenties to fifties showed a partial correlation with COVID-19, having a partial correlation with tourist spots, which was correlated with restaurants and variables of mobility?” Besides just commenting on the graphs I think the authors should also make an effort to provide some interpretation behind this or use additional plots to help the reader follow the logic.

8) L334, “we did not take the intercorrelation among the samples (prefectures) into account when constructing the graphical model” How can this affect the results?

9) Among the limitations, also the possible biases coming from the datasets (especially mobility ones) used should be discussed

Reviewer #2: The study investigates the effects of the state of emergency and go-to travel policies on the number of COVID-19 cases in Japan. This study concludes that the state of emergency affected the control of COVID-19 spread and that the go-to campaign increased the number of cases by changing the locations of social contacts. The objective of the study is well motivated -- to provide evidence for lockdown policies based on big mobility data, which could be useful for policy makers. However, the statistical analysis (which the methods are very unclear) are very weak and lacks the rigor to reach any conclusions on the impact of the policies on the number of infections.

Here are some comments that I recommend the authors to address:

- The methodological procedure of the analysis is very hard to follow. I strongly recommend the authors to provide more detail on the statistical analysis method they used to produce the network plots. Are we looking at just correlations between variables? or are we able to capture any causal structures between them?

- page 17 line 241 what does this mean? "The directed edge (-) from the variables in the group of factors to the variables in the group of outcomes indicates the time order." How can we tell the 'time order'? and what do you mean by 'time order'?

- Figure 2 & 3 are visually very hard to read. Please update using more high resolution figures.

- Overall, the analysis is very weak (merely captures time-lagged partial correlation only?) and thus the discussions/conclusions are not convincing. At least some econometric analysis that can give statistical conclusions on the effects of covariates on the dependent variable is needed; there are already many studies that use epidemiological models (e.g., SIR, SEIR) to rigorously achieve these kinds of analysis, which may be a path that the authors could pursue.

6. PLOS authors have the option to publish the peer review history of their article (what does this mean?). If published, this will include your full peer review and any attached files.

Reviewer #1: No

Reviewer #2: No

---

## [Author Response · Author response to Decision Letter 0]

10 Feb 2022

Thank you for the careful review of our manuscript.

We have considered the helpful comments from the reviewer and the editor, and we have revised the manuscript in response to these comments. The changes made are highlighted in the manuscript, and a detailed point-by-point response is described in the document " Respons to Reviewers".

We hope that, with the revisions described herein, the article is now acceptable for publication in PLOS ONE.

Thank you for taking the time to consider the manuscript. I look forward to hearing from you soon.

---

## [Decision Letter · Decision Letter 1]

8 Apr 2022

Changes in social environment due to the state of emergency and Go To campaign during the COVID-19 pandemic in Japan: an ecological study

PONE-D-21-31460R1

Dear Dr. Nojiri,

We’re pleased to inform you that your manuscript has been judged scientifically suitable for publication and will be formally accepted for publication once it meets all outstanding technical requirements.

Kind regards,

Michele Tizzoni

Academic Editor

PLOS ONE

Additional Editor Comments (optional):

Reviewers' comments:

Reviewer's Responses to Questions

**Comments to the Author**

1. If the authors have adequately addressed your comments raised in a previous round of review and you feel that this manuscript is now acceptable for publication, you may indicate that here to bypass the “Comments to the Author” section, enter your conflict of interest statement in the “Confidential to Editor” section, and submit your "Accept" recommendation.

Reviewer #1: All comments have been addressed

Reviewer #2: All comments have been addressed

2. Is the manuscript technically sound, and do the data support the conclusions?

Reviewer #1: Yes

Reviewer #2: Yes

3. Has the statistical analysis been performed appropriately and rigorously? 

Reviewer #1: Yes

Reviewer #2: Yes

4. Have the authors made all data underlying the findings in their manuscript fully available?

Reviewer #1: Yes

Reviewer #2: No

5. Is the manuscript presented in an intelligible fashion and written in standard English?

Reviewer #1: Yes

Reviewer #2: Yes

6. Review Comments to the Author

Reviewer #1: (No Response)

Reviewer #2: I believe all my comments have been addressed. The formatting of the manuscript is different from the standard PLoS ONE articles (Introduction, Results, Discussion, Materials and Methods) so it may be worth changing the structure of the paper, but I leave the decision to the editor.

7. PLOS authors have the option to publish the peer review history of their article (what does this mean?). If published, this will include your full peer review and any attached files.

Reviewer #1: No

Reviewer #2: No

---

## [Editor Report · Acceptance letter]

19 Apr 2022

PONE-D-21-31460R1 

Changes in social environment due to the state of emergency and Go To campaign during the COVID-19 pandemic in Japan: an ecological study 

Dear Dr. Nojiri:

I'm pleased to inform you that your manuscript has been deemed suitable for publication in PLOS ONE. Congratulations! Your manuscript is now with our production department. 

Kind regards, 

on behalf of

Dr. Michele Tizzoni 

Academic Editor

PLOS ONE